# Specific Emitter Identification Based on Self-Supervised Contrast Learning

Bo Liu, Hongyi Yu *, Jianping Du 🆔, You Wu, Yongbin Li, Zhaorui Zhu and Zhenyu Wang

Information System Engineering College, PLA Strategic Support Force Information Engineering University, Zhengzhou 450001, China
* Correspondence: xxgcmaxyu@163.com

**Abstract:** The current deep learning (DL)-based Specific Emitter Identification (SEI) methods rely heavily on the training of massive labeled data during the training process. However, the lack of labeled data in a real application would lead to a decrease in the method's identification performance. In this paper, we propose a self-supervised method via contrast learning (SSCL), which is used to extract fingerprint features from unlabeled data. The proposed method uses large amounts of unlabeled data to constitute positive and negative pairs by designing a composition of data augmentation operations for emitter signals. Then, the pairs would be input into the neural network (NN) for feature extraction, and a contrastive loss function is introduced to drive the network to measure the similarity among data. Finally, the identification model can be completed by fixing the parameters of the feature extraction network and fine-tuning with few labeled data. The simulation experiment result shows that, after being fine-tuned, the proposed method can effectively extract fingerprint features. When the SNR is 20 dB, the identification accuracy reaches 94.45%, which is better than the current mainstream DL approaches.

**Keywords:** Specific Emitter Identification; self-supervised contrast learning; data augmentation

## 1. Introduction

Specific Emitter Identification aims to recognize different emitters by extracting the radio frequency features from the signal. The features are as difficult to forge as human fingerprints, and they are thus also called "fingerprint features" of the emitter. The traditional SEI methods originate from human's observation on signal appearance characteristics and specification parameters, such as carrier frequency, frequency modulation angle and so on. In recent years, scholars tried to analyze the signal characteristics using mathematical tools. They explored the generation mechanism of the traits, and they designed methods such as higher-order statistics [1], domain transformation [2], model parameter estimation, etc. The signal analysis-based methods mainly deal with signals in the time domain, frequency domain, or time–frequency domain, and have achieved good performance. Although the signal analysis-based algorithms are highly interpretable, they are still limited by the following shortcomings: first, prior knowledge is required for artificial feature extraction, and the process is labor intensive and time consuming; second, the parameters and threshold of the theoretical model need to be manually designed, which would affect the final versatility of the method.

With the rapid development of DL, its powerful automatic feature extraction capability provides us with new ideas for SEI. These identification methods based on DL usually combine artificial intelligence with signal processing, and most of them adopt the step of feature transformation—data input—network training—identification [3–5]. For example, I/Q complex baseband could be directly sent to the neural network for feature extraction [6], and the identification can be completed through the original information. HOU et al. used the Deep Convolutional Neural Network (DCNN) and random forest at the same time,

generating a comprehensive weight vector for identification [7]. DING et al. designed the method of using high-order spectrum as input data, which suppresses the negative impact of noise by converting the signal into bispectrum and achieves an accuracy of 87% over five emitters [8]. Based on Generative Adversarial Networks (GANs), Gong et al. proposed an Information Maximizing Generative Adversarial Network (InfoGAN) [9]. The bispectral images extracted from the received signal are converted into grayscale histogram as input data. NIU et al. also proposed a PACGAN (Pooling Auxiliary Classifier Generative Adversarial Network) algorithm for SEI tasks, in which the input signal is converted into the differential constellation trace figure [10].

Although the above methods achieve good performance, these studies are mainly based on supervised learning, which requires large amounts of labeled data. If there is insufficient labeled data, the deep neural network is prone to suffer the problem of overfitting. Thus, the final result would be affected seriously. While the vast majority of SEI tasks aim to identify unfamiliar signals, labeling the rich unlabeled data resources manually in practice is very expensive.

Therefore, it is particularly necessary to study the problem of Specific Emitter Identification when only a small amount of unlabeled data could be used. Model pre-training, which is viewed as an effective method using unlabeled data, has achieved great success in recent years for tasks such as computer vision (CV) and natural language processing (NLP). Oord et al. proposed a contrast predictive coding model for text, audio and other temporal data, in which pre-training was achieved by segmenting the samples and comparing the mutual information between the feature coding of the segments of data [11]; On the basis of this, a dynamic dictionary is constructed using a momentum updating encoder based on the contrast loss function to improve the effectiveness of model pre-training [12,13], and a simple framework for contrastive learning of visual representations (SimCLR) was proposed in [14–16] to provide self-supervised information for model training through data augmentation, reducing the reliance on labelled samples for image classification tasks. The above models are all based on the idea of self-supervised learning for model pre-training, and are widely used in tasks such as image recognition and machine translation, but less in signal processing. Aiming at the above problems, an SEI method SSCL based on self-supervised contrast learning is proposed [17–20]. Self-supervised contrast learning exploits the similarity between sample pairs to mine the feature representation from large amounts of unlabeled data. It is an effective way to achieve a pre-trained model using unsupervised learning approaches. Recent studies have shown that, compared with current supervised learning methods, a pre-trained self-supervised contrast learning model only needs few labeled data for fine-tuning to obtain a similar performance [14]. In addition, in view of the subtle characteristics of the SEI fingerprint feature, SSCL chooses constellation trace figures as the input data, which suppresses the interference of the symbol information, and a stochastic data augmentation module suitable for an emitter signal is designed to meet the requirement of constituting positive and negative pairs needed in contrast learning.

The main contributions of this paper are as follows:

(1) Based on contrastive learning, we propose a self-supervised method for SEI tasks. The proposed method can fully use large amounts of unlabeled samples for model pre-training, effectively extract the fingerprint features of the emitter signal automatically and improve generalization ability.

(2) Considering the stability of the fingerprint feature in the time domain, a stochastic data augmentation module suitable for emitter signals is designed. As a process of increasing the number and diversity of samples, data augmentation has achieved great progress in the training of DL networks for CV tasks. However, it has not been widely used in the field of SEI. The module proposed can effectively constitute positive and negative sample pairs using unlabeled data, which helps to establish an identification model using contrast learning.

The structure of this paper is organized as follows. The signal model we use for SEI is introduced in the second part. The third part is the introduction of the proposed SSCL

method. The fourth part presents and discusses the case studies. In the final part, the paper is summarized.

## 2. Signal Model

In this paper, we consider that the emitter signal with fingerprint features would be affected by additive noise in the channel transmission process, assuming that there is no other influence. Thus, we can express the received signal as:

$$r(t) = s(t) + n(t), \tag{1}$$

where $r(t)$ represents the signal we actually received, $s(t)$ represents the signal carrying the transmitter hardware distortion information, and $n(t)$ represents the addictive Gaussian white noise existing in the channel.

Due to the limitation of the manufacturing level, there would be a deviation between the actual and the ideal characteristics of various electronic components of the transmitter, resulting in some subtle distortion of the transmitted signal. For different transmitters, this kind of distortion would be different and keep stable for a long time. So, it can be used as a fingerprint feature to identify different emitters. The signal carrying fingerprint features can be expressed as:

$$s_i(t) = f_i(s_{0i}(t)), i = 1, \dots, n, \tag{2}$$

where $s_i(t)$ represents the transmit signal of the $i$th emitter, $n$ represents the number of different emitters, $s_{0i}(t)$ represents the ideal baseband signal, and $f_i$ represents the modulation process, including both the intentional modulation that generates symbol information and the unintentional modulation that generates fingerprint features.

This experiment focuses on the IQ modulator that can generate fingerprint features in the transmitter, and builds a signal generation model. Assuming an ideal baseband signal

$$s_0(t) = s_{b,I}(t) + js_{b,Q}(t), \tag{3}$$

where $s_{b,I}(t)$ and $s_{b,Q}(t)$ are real parts and imaginary parts of the time-varying baseband signals, respectively. Consider an IQ modulator signal that has no distortion:

$$s(t) = s_{b,I}(t) \cos(2\pi f_c t) - s_{b,Q}(t) \sin(2\pi f_c t), \tag{4}$$

where $f_c$ is the carrier frequency. Then, the baseband signal with an IQ imbalance would be modulated as:

$$s(t) = (1 - g)(s_{b,I}(t) + c_I) \cos(2\pi f_c t) - (1 + g)(s_{b,Q}(t) + c_Q) \sin(2\pi f_c t + \varphi), \tag{5}$$

where $g$ is the gain imbalance of the transmitter, $c_I$ and $c_Q$ represent the DC offset and $\varphi$ is the phase imbalance.

## 3. Method

The excellent performance of supervised learning cannot be achieved without expensive labeled datasets. Therefore, unsupervised learning is studied by scholars as an alternative to avoiding the use of large-scale annotated datasets. As a representative branch, self-supervised learning can help the model learn feature representations by designing downstream tasks and using the input data itself as supervision. Commonly, self-supervised learning can be summarized as follows: generative and discriminative.

The traditional generative approaches learn representation through training an encoder to map the input data $x$ to a hidden vector $h$, then reconstruct the input data $x$ through the decoder and calculate the reconstruction loss. The representative method is the autoencoder (AE) [21] proposed in 1987, and in 2014, the emergence of the Generative Adversarial Network (GAN) made the generative methods become popular again and they continued to develop.

For a long time, the generative approaches have been considered the only option for self-supervised ways. However, discriminative approaches represented by contrast learning have recently revealed huge potential in many fields, such as CV, NLP, etc. Like supervised learning methods, contrast learning also uses objective functions to learn representations, but trains the network with inputs from unlabeled datasets. Contrast learning aims to bring those similar samples closer together and different samples farther away. Therefore, it uses a similarity metric to measure the distance between two embeddings.

SimCLR, proposed by Chen Ting et al. [14], is a general framework of self-supervised contrast learning for CV tasks. Compared with others, it has a simpler implementation, better performance and better versatility. Inspired by SimCLR, we designed an SEI method SSCL based on self-supervised contrast learning. By constituting positive and negative pairs and using massive unlabeled data for a pre-trained model, the fingerprint features can be extracted from emitter signals effetely and automatically. The framework of SSCL is shown in Figure 1.

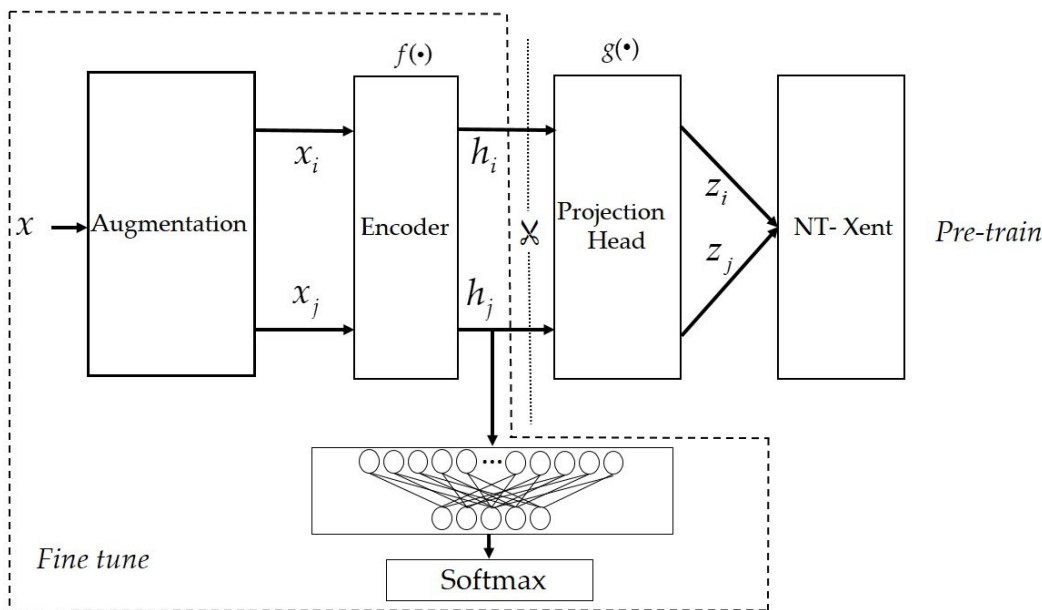

**Figure 1.** The framework of SSCL.

The training of SSCL includes a pre-training process and a fine-tuning process. The pre-training process uses a large number of unlabeled data to train the feature extraction network (the Encoder and the Projection Head) parameters. First, unlabeled emitter signals are input to the stochastic data augmentation module. After a series of data preprocessing operations, two constellation trace figures are obtained as positive pairs. Then, they are input to the Encoder for feature extraction. The extracted features pass through the Projection Head to obtain the feature projection in the hypersphere space [22]. Lastly, the parameters of the feature extraction network are updated under the action of the loss function to learn a better feature projection distribution.

The fine-tuning process actually trains a simple classifier using few labeled data. The Encoder parameters are frozen, and the Projection Head is replaced by a two-layer fully connected network. Therefore, only a small number of parameters need to be updated in the fine-tuning process. The training can be completed by using the cross-entropy loss function with few labeled data. Finally, the fine-tuned model can be directly used to identify emitter signals.

The framework of SSCL is mainly composed of three parts: the stochastic data augmentation module, feature extraction network, and contrastive loss function.

### 3.1. Stochastic Data Augmentation

The stochastic data augmentation module is a composition of a series of data preprocessing operations. For contrast learning, the samples used in model pre-training are all unlabeled data, so the information learned only comes from the samples themselves. As shown in Figure 2, when the original sample is passed into the module, two augmented samples with the same category but different contents are generated, constituting a positive sample pair. For comparison, the augmented samples generated from other original data in the same training batch are uniformly considered as negative sample pairs.

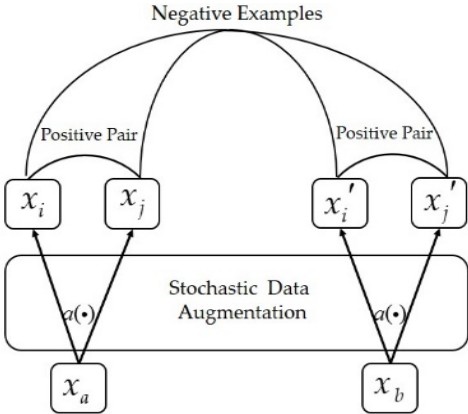

**Figure 2.** Negative pair and Positive pair.

So the two augmented samples in the positive sample pair are derived from the same original data and are similar to each other; the two augmented samples in the negative sample pair are derived from different original data and are different from each other. Contrast learning just learns representations with classification ability from the similarity and difference between samples. Therefore, the key of SSCL is to correctly define the operations composition of stochastic data augmentation.

However, most of the current stochastic data augmentation methods are only suitable for CV tasks. Their application in terms of signals only consists of several operations, such as noise addition, etc. [23]. To use contrast learning for SEI, it is necessary to design operations suitable for the emitter signals. In this regard, there are two main considerations:

(1) Convert emitter signals to images. Thus, we can draw on the data augmentation method for CV tasks. The work of [24,25] shows that constellation trace figures could well preserve the fingerprint feature. However, what cannot be ignored is that the fingerprint features are relatively subtle. So operations such as cropping, scaling, and masking commonly used for CV tasks would destroy the original information. The trained model would collapse and be led to failure. Thus, these operations need to be discarded.

(2) The stochastic data augmentation aims to use original samples to constitute positive pairs belonging to the same class. Then, the feature extraction network is forced to pay attention to the invariance of the features of the sample data in multiple disturbances. As the previous section pointed out, the fingerprint feature of the emitter signal could keep stability in the time domain. If the signal is truncated at the midpoint, its symbol information carried by the two parts would be different, but the fingerprint features would be the same. Therefore, considering the two truncated samples as a positive pair is undoubtedly suitable for the actual characteristics of the signal.

Based on these considerations, we propose a stochastic data augmentation method suitable for emitter signals in SSCL. When a signal has a random disturbance, the fingerprint features are retained as much as possible. The operation composition and parameters setting are shown in Table 1. Note that when a fixed-length emitter signal passes through the stochastic data augmentation module, its output will vary with the model training

process. In pre-training, the output of the module is two constellation trace figures that form one positive sample pair, which should be sent to the Encoder at the same time. Whereas, in the fine-tuning and testing process, due to the non-necessity of constituting positive and negative pairs, the two output figures can be seen as two unrelated samples, participating in the next fine-tuning or testing process.

**Table 1.** Stochastic data augmentation and parameters setting.

| Step | Operation | Parameter | Probability |
| --- | --- | --- | --- |
| 1 | Unified Length | 2000 symbols | — |
| 2 | Truncation | — | — |
| 3 | Convert to Image | Size (224, 224) | — |
| 4 | Grayscale | Channels = 1 | — |
| 5 | Random Color Distortion | — | 0.8 |
| 6 | Random Horizontal Flip | — | 0.5 |
| 7 | Random Vertical Flip | — | 0.5 |

*3.2. Feature Extraction Network*

In most DL models, NNs are used to learn representations from samples. Most of the current contrast learning methods [14,19,21] adopt this approach.

The feature extraction network of SSCL follows the design of the Encoder and the Projection Head in SimCLR. The Encoder aims to extract effective feature vectors from input samples, which are generally implemented by classical networks, such as the Convolutional Neural Network (CNN), Transformer, etc. Considering the characteristics of the emitter signal, the number of samples, and the training cost, we choose the ResNet [26] with strong representation ability as the Encoder of SSCL. ResNet uses BasicBlock with the shortcut connection structure to alleviate the degradation problem when NN has too many layers to be trained. The ResNet used by SSCL consists of eight BasicBlocks with the same structure but different parameters. The specific framework is shown in Figure 3.

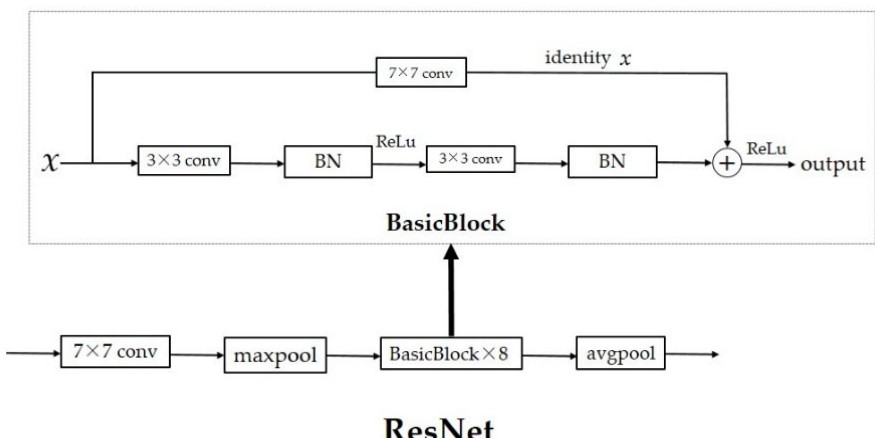

**Figure 3.** The framework of Encoder.

The Projection Head uses a two-layer Multilayer Perceptron (MLP), mapping the encoded sample features through the nonlinear operation into the hypersphere space. Then, the contrastive loss function is used to measure the similarity and difference between sample features. Ref. [14] shows that, when the Projection Head adopts fully connected networks with nonlinear excitation, the model performance would be significantly improved. Table 2 gives the specific parameters about the input data format of the feature extraction network and the dimension change after passing through each layer of the network.

**Table 2.** SSCL network parameters.

|  | Module Name | Output Size |
|---|---|---|
|  | Input data | $224 \times 224 \times 1$ |
| Encoder | $7 \times 7$ conv | $112 \times 112 \times 64$ |
|  | Maxpool | $56 \times 56 \times 64$ |
|  | BasicBlock 1 | $56 \times 56 \times 64$ |
|  | BasicBlock 2 | $56 \times 56 \times 64$ |
|  | BasicBlock 3 | $28 \times 28 \times 128$ |
|  | BasicBlock 4 | $28 \times 28 \times 128$ |
|  | BasicBlock 5 | $14 \times 14 \times 256$ |
|  | BasicBlock 6 | $14 \times 14 \times 256$ |
|  | BasicBlock 7 | $7 \times 7 \times 512$ |
|  | BasicBlock 8 | $7 \times 7 \times 512$ |
|  | Avgpool | $1 \times 1 \times 512$ |
| Projection Head | Perceptron | 256 |
|  | Perceptron | 128 |

*3.3. Contrastive Loss Function*

The contrastive loss function is used to measure the similarity measurement error between samples in the model pre-training process. A minibatch of $N$ examples is randomly sampled and then is input into the augmentation module, resulting in $2N$ data points to form a set $z = \{z_1, z_2, \ldots, z_{2N-1}, z_{2N}\}$. Given a positive pair $z_i$ and $z_i^+$, we treat the other $2N - 2$ augmented examples as negative examples within a minibatch. The contrastive loss function aims to help samples identify their positive pair in a minibatch by improving the similarity between positive sample pairs and reducing the similarity between negative pairs.

To evaluate this similarity loss, SSCL uses the normalized temperature-scaled cross-entropy loss (NT-Xent) in [11,19,27] as the loss function. NT-Xent is defined as follows:

$$\mathcal{L} = \frac{1}{2N} \sum_{k=1}^{N} [\ell(2k-1, 2k) + \ell(2k, 2k-1)], \tag{6}$$

where $2k - 1$ and $2k$ is the serial number of the positive sample pair in the batch, $N$ is the training batch size and $\ell(i, j)$ is the loss function of the positive pair:

$$\ell(i, j) = -\log \frac{\exp(\text{sim}(z_i, z_j)/\tau)}{\sum\limits_{k=1}^{2N} \mathbb{1}_{[k \neq i]} \exp(\text{sim}(z_i, z_k)/\tau)} \tag{7}$$

where $\mathbb{1}_{[k \neq i]}$ is an indicator function evaluating to 1 if $k \neq i$ else 0, $z$ represents the mapping feature generated by the Projection Head and $\tau$ represents a hyperparameter. $\text{sim}(\bullet)$ is the sample similarity function, which is defined as follows:

$$\text{sim}(z_i, z_j) = \frac{z_i^T z_j}{\|z_i\| \|z_j\|}, \tag{8}$$

In particular, due to the size of the training batch and the class number of emitter signals, samples originally belonging to the same class in each batch are inevitably regarded as negative sample pairs. In this way, the contrastive loss function would output penalty factors with different weights according to the similarity between samples. These factors determine the intensity of the distribution change of the sample feature projection in the hypersphere space during each training. Given a random positive sample pair, the negative samples that belong to the same class would be less repelled than other negative samples due to their higher similarity.

*3.4. Training Strategy*

The training process of SSCL can be summarized in two parts:

The pre-training process is shown in Algorithm 1. A minibatch $\{x_k\}_{k=I}^N$ is sampled from the training set $D_t = \{x_i\}$, through the stochastic data augmentation module, generating augmented samples. In the same batch of training samples, a total of $N$ positive sample pairs and $2N(N-1)$ negative sample pairs are generated. Then, the feature projections are obtained in the hypersphere space through the feature extraction network. Finally, the feature projections are calculated by the NT-Xent loss function in Equation (6). By influencing the distance between positive and negative sample pairs, the model is forced to learn the discriminative representations of the sample data, and complete the model pre-training process.

---

**Algorithm 1.** The pre-train learning algorithm of SSCL.

---

**Input:** Structure of $f(\bullet)$, $g(\bullet)$, constant $\tau$, and batch size $N$
**for** sampled minibatch $\{x_i\}_{i=0}^N$ **do**
    **for all** $\{x_1, x_2, x_3, \ldots, x_N\}$ do
        *#draw two augmented Operations*
        $x_{2k-1}^{\text{Aug}} = \text{Aug}_1(x_i)$      $x_{2k}^{\text{Aug}} = \text{Aug}_2(x_i)$      #Augmentation
        $h_{2k-1} = f(x_{2k-1}^{\text{Aug}})$      $h_{2k} = f(x_{2k}^{\text{Aug}})$      #Encoder
        $z_{2k-1} = f(h_{2k-1})$      $z_{2k} = f(h_{2k})$      #Projection
    **end for**
    **for all** $i, j \in \{1, \ldots, 2N\}$ **do**
        $\text{sim}(z_i, z_j) = z_I^T z_j / (||z_i|| \cdot ||z_j||)$ #pairwise similarity
    **end for**
    **define** $\ell(i, j)$ and L
    update the parameters of $f(\bullet)$ and $g(\bullet)$ to minimize L
**end for**
output networks $f(\bullet)$ and remove $g(\bullet)$

---

The fine-tuning process is to remove the Projection Head after pre-training, and then construct a model for SEI by connecting a fully connected layer with the Softmax classifier. The function of the fully connected layer is to perform a linear operation on the feature vector extracted by the Encoder, and realize the conversion from the feature dimension to the classifier dimension without losing any information. When training for the newly added fully connected layer, the number of parameters to be learned is tiny. So, only a few labeled samples are needed. At last, the fine-tuned model can be used to identify or test emitter signals.

## 4. Experiment and Discussion

To test the validity of SCLL, we conduct experimental verification on the simulation dataset. All the experiments are conducted under the environment of PyTorch 1.10 (windows), and run on a computer with an Intel core i7—11700 k processor, NVIDIA RTX 2080Ti, and 32 GB RAM.

*4.1. Experiment Setup*

Using the signal model in Section 2, the signal in the experiment is modulated by QPSK, and each signal sample contains 2000 symbols. After being truncated at the midpoint, each sample contains 1000 symbols. The symbol rate is 100 K Baud, the carrier frequency is 350 kHz, and the sampling rate is 1 MHz. The baseband shaping filter uses a raised cosine pulse with a roll-off factor of 0.35.

Then, the IQ modulation simulation signal with distortion is converted into a constellation trace figure as the sample input form. Five types of emitter signals with different distortions are set up in the experiment, and the distortion parameter setting is referred

to [28], as shown in Table 3. The constellation trace figure we use as input data is generated, as shown in Figure 4.

**Table 3.** Parameter settings for different specific emitters.

| Parameter | $g$ | $\varphi$ | $c_I$ | $c_Q$ |
|---|---|---|---|---|
| 1 | 0.0299 | 0.0137 | 0.0142 | 0.0147 |
| 2 | 0.0188 | 0.0093 | 0.0097 | 0.0102 |
| 3 | 0.0081 | 0.0050 | 0.0052 | 0.0057 |
| 4 | −0.0025 | 0.0006 | 0.0007 | 0.0012 |
| 5 | −0.0128 | −0.0038 | −0.0038 | −0.0033 |

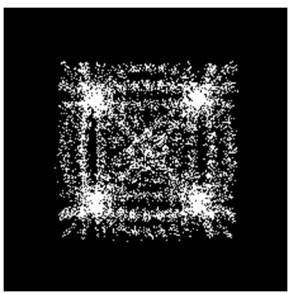 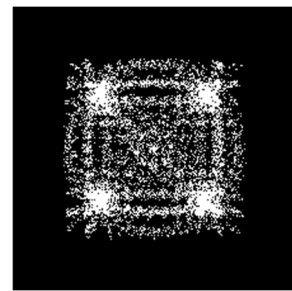

**Figure 4.** Constellation trace figures.

In the experiment, samples under different SNR levels in the range of {0, 5, 10, 15, 20} dB are selected for model training and evaluation. The number of samples for each type of emitter signal under each SNR is 1250, of which 1000 are used for pre-training and 250 are used as a test dataset.

As listed in Table 4, we used the Adam optimizer in training; the basic learning rate is $10^{-3}$, the weight decay is set to $10^{-6}$, and the batch size we use is 128. In addition, 100 epochs are trained in each training process.

**Table 4.** SSCL model parameters.

| Hyper Parameter | Pre-Train | Fine Tune |
|---|---|---|
| Batch size | 128 | 256 |
| Learning rate $l_r$ | $10^{-3}$ | $10^{-3}$ |
| Weight decay | $10^{-6}$ | $10^{-6}$ |
| Max epochs | 100 | 100 |
| Embedding dim $d$ | 512 | — |
| Temperature $\tau$ | 0.7 | — |

*4.2. Results and Comparison*

4.2.1. Performance of the Model with Few Labeled Data

Three classical DL methods are selected as comparison, including ResNet, the same residual network with using SSCL, and we train it as a supervised learning method by the cross-entropy loss function; CAE [29], the convolutional autoencoder based on the self-supervised learning; CNN [30], the convolutional neural network based on the supervised learning. To make each model suitable for the dataset, the model structure is adjusted appropriately.

We randomly sampled 10 annotated samples from the training set to train CNN, ResNet, CAE, and SSCL, respectively. Among them, CAE and SSCL need to use the unlabeled sample training set to pre-train first, and then connect a fully connected (FC) layer and Softmax classifier for supervised fine-tuning.

Figure 5 shows the identification accuracy of the four models versus SNRs. In the case of few labeled data, CNN and ResNet trained by supervised learning cannot be fully

trained. The identification accuracy under 15 dB is only 42.72% and 29.16%, respectively, reflecting the dependence of supervised deep learning approaches on large-scale labeled datasets. CAE, based on the generative self-supervised method, extracts the key features reflecting the global information in the sample through the reconstruction loss. Still, it does not show good identification performance, and the accuracy is 43.60% under 15 dB, similar to CNN. Exactly what happened is that the fingerprint feature of the emitter signals is highly susceptible to interference by other information of the signal or image due to its subtle characteristics. Therefore, the features extracted by the reconstruction loss are not distinguishable enough, even though they reflect the global information based on removing redundant information.

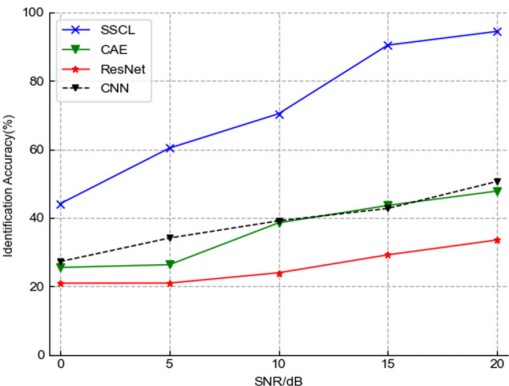

**Figure 5.** Identification accuracy under different models.

In contrast, SSCL can obtain acceptable results under few labeled samples, with the identification accuracy reaching 90.45% when the SNR is 15 dB. This is because the discriminative features of unlabeled samples are extracted by constituting positive and negative sample pairs in the pre-training process, which proves the effectiveness of the SSCL method.

### 4.2.2. The Effect of Pre-Training

To observe the effect of pre-training on the model performance more intuitively, we use the t-SNE to reduce the dimension of the output features. Then, we can see the distribution of the output features of the Encoder. In the experiment, 100 examples of five classes are randomly generated by the same model under 15 dB, which are input to the Encoder before and after pre-training. Figure 6 shows the identification accuracy changes of SSCL. The pre-trained SSCL is significantly higher than the one without pre-training. When the SNR is 0, 5, 10, 15, and 20 dB, the leading ranges are 23.48%, 38.88%, 35.84%, 55.65%, and 57.55%, respectively.

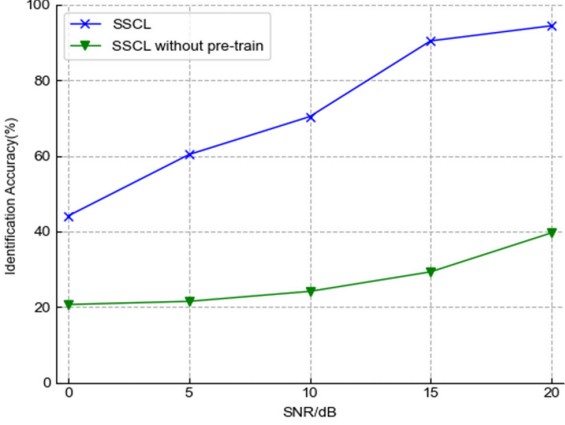

**Figure 6.** Identification accuracy with or without pre-training.

The output features visualizations of the Encoder are shown in Figure 7. It can be seen that after the Encoder is pre-trained, the distribution of different classes of samples shows apparent discrepancies. That proves that the output features after pre-training can be segmented by a linear classifier in the following process of fine-tuning, which is also confirmed by Experiment 1. The results show that SSCL can effectively extract the potential discriminative features of unlabeled samples by using contrastive loss for pre-training.

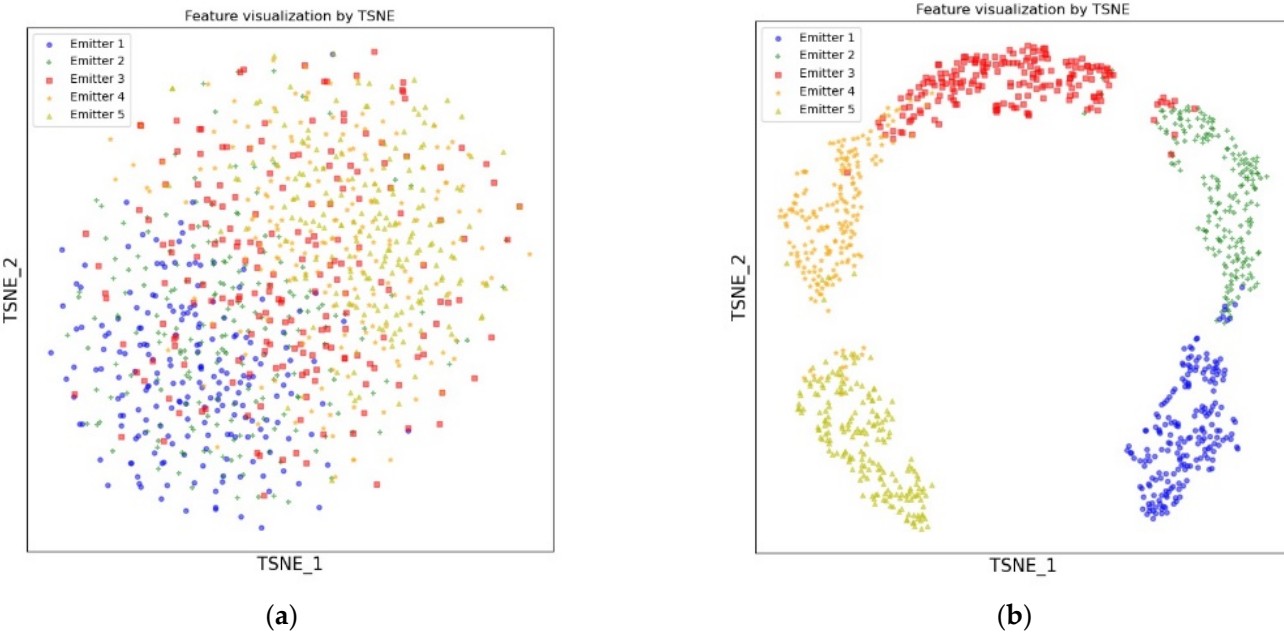

(**a**)          (**b**)

**Figure 7.** T-SNE visualizations. (**a**) The output of Encoder without pre-training; (**b**) The output of Encoder with pre-training.

### 4.2.3. The Effect of Stochastic Data Augmentation

Section 3 points out that, different from the image task, the operation of truncating the signal to constitute positive pairs is crucial to data augmentation operations in SSCL. Before and after SSCL removes the image data augmentation and signal truncation operations, Figure 8 shows the change in model performance, and Figure 9 shows the output features visualizations of the Encoder under 20 dB.

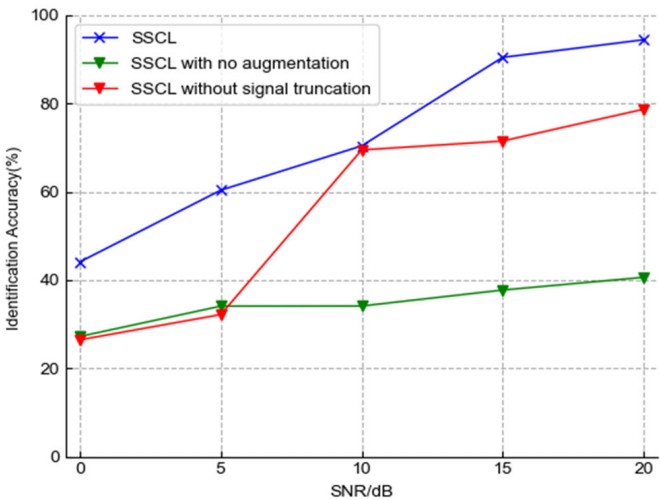

**Figure 8.** Identification accuracy with different data augmentation operations.

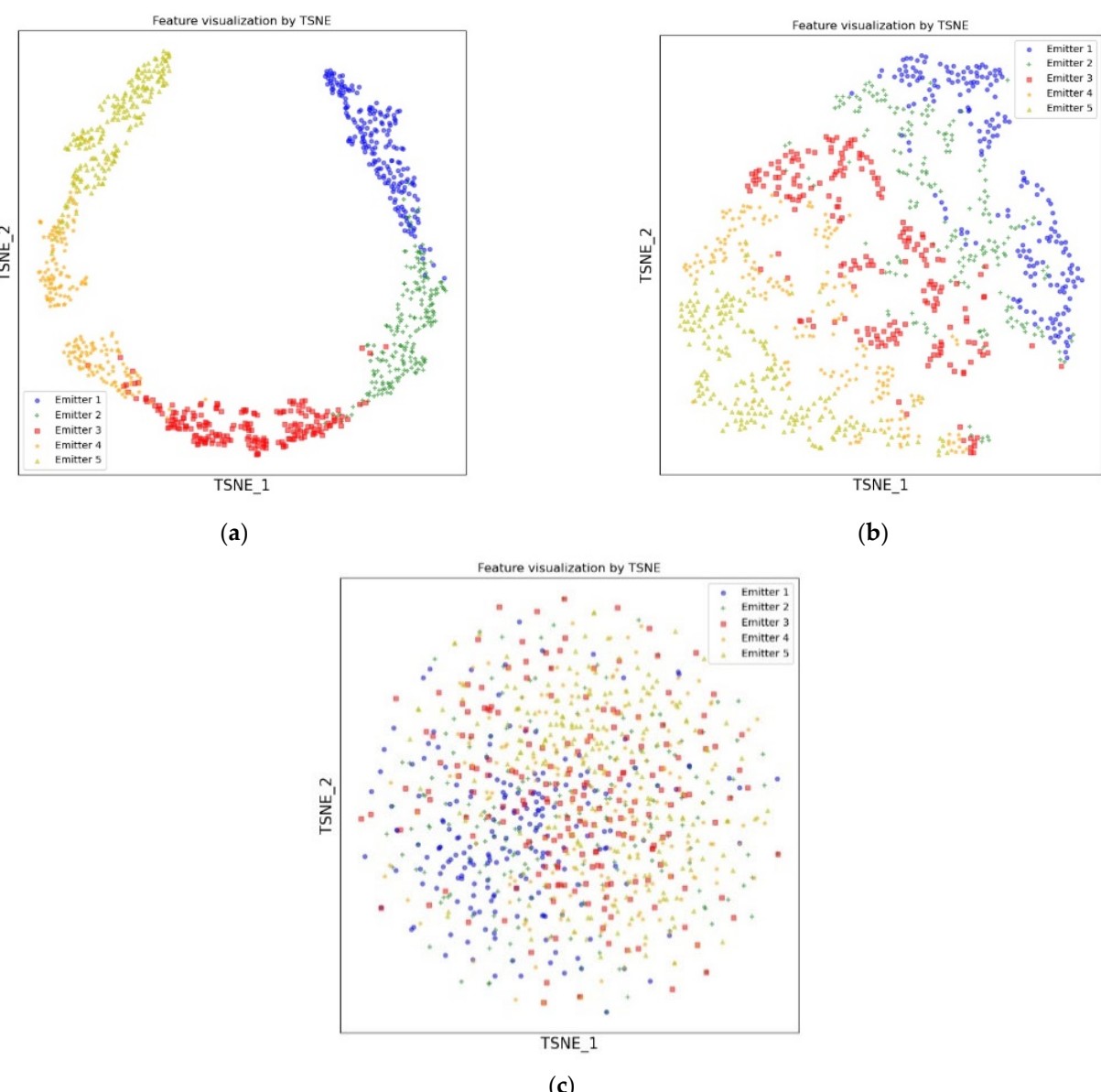

**Figure 9.** T-SNE visualizations. (**a**) The output of Encoder with image data augmentation and signal truncation; (**b**) The output of Encoder with only image data augmentation; (**c**) The output of Encoder with no augmentation.

We can see that the model of directly removing the stochastic data augmentation module still uses the contrastive loss for pre-training. However, the identification accuracy is only 36.90%, and the output feature visualizations of the Encoder are also very chaotic, not reflecting a good classification potential.

When the model only uses the image data augmentation operations, the identification accuracy rate reaches 78.70% under 20 dB, and the feature visualizations of different samples also produce significant discrepancies. As the model adopts the signal truncation operation, the identification accuracy is greatly improved under the four SNR environments of 0 dB, 5 dB, 15 dB, and 20 dB. The leading amplitudes are 17.68%, 28.20%, 18.95%, and 15.75%, respectively. The feature distribution of the Encoder not only has obvious discrimination between classes, but also has a great improvement in the clustering degree within the class.

The above results prove that the stochastic data augmentation operations designed for emitter signals in SSCL is suitable for specific task scenarios and plays a vital role in the feature extraction of unlabeled data.

## 5. Conclusions

Facing the contradiction between the vast demand for data-driven deep learning technology and the expensive cost of manually labeled data, unsupervised or semi-supervised learning methods have become a research hotspot for SEI. This paper proposes a novel method (SSCL) based on self-supervised contrast learning, and experimental research and demonstration are carried out.

We summarize this paper as follows. Based on the stability of the fingerprint feature of the emitter signal in the time domain, the data augmentation operations using signal truncation and image conversion successfully constitute the positive and negative pairs. Those pairs play an important role in the feature extraction of unlabeled data. By constructing positive and negative pairs to pre-train the model with unlabeled data, SSCL achieves better classification performance only using a small amount of labeled data, which provides a new research idea for SEI.

However, inevitably, SSCL has some problems, and we list the limitations and outlook. First, the network architecture still has many hyperparameters that needed to be manually tried and determined with prior knowledge, which we will work on to improve. Second, SSCL has good identification performance under high SNR, but its advantage is not apparent under low SNR, which limits the application in practical tasks. Based on SSCL, we will further explore methods to improve its performance under common SNR conditions.

**Author Contributions:** Conceptualization, B.L.; formal analysis, B.L.; writing, B.L.; formal analysis, H.Y.; project administration, H.Y. and J.D.; data curation, Y.W. and Y.L.; validation, Z.Z. and Z.W. All authors have read and agreed to the published version of the manuscript.

**Funding:** This research received no external funding.

**Data Availability Statement:** The data used in this paper can be obtained by contacting the corresponding author.

**Conflicts of Interest:** The authors declare no conflict of interest.

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
