# Peer review of "Specific Emitter Identification Based on Self-Supervised Contrast Learning"

_electronics, doi:10.3390/electronics11182907_

Round 1

Reviewer 1 Report

The title and abstract are acceptable. The Siamese network as a tool for a self-supervised network is only referred in the abstract. The other types of networks should be explained with a focus on Siamese network architecture. this issue can be resolved by some rearrangement in the text.

please check the structure of the references. Most of the figures(such as Fig 9) need more clarification considering axis info, parameter names, and configs.

Actually, I can imagine the benefits of the proposed method. the paper needs to focus on the hardness of the problem tackled in the manuscript. a section on the problem statement and elaborating on the connections between the problem and other NP-Hard problems may be useful.

Reviewer 2 Report

Specific Emitter Identification Based on Self-supervised Contrast Learning

The authors of this paper propose a self-supervised method via contrast learning (SSCL), which is used to extract fingerprint features from unlabeled data. The simulation experiment result shows that after fine-tuned, the proposed method can effectively extract the fingerprint features.

Comments/suggestions

-          A separate literature review session should be added, after Introduction. More new references should be added

-          The results must be presented clearly

-          The research design is appropriate

-          The methods are adequately described

-          Conclusions should be supported by the results

-          Editing for English language is required

Round 2

Reviewer 1 Report

actually, I got most of my answers during revision.

my main concern is images. the quality of images is low and sometimes the reader may consider that they are copied from other sources. Please consider this issue during the preparation of the final version. a table of symbols is also recommended.